# Smart Biomimetic Nanozymes for Precise Molecular Imaging: Application and Challenges

**DOI:** 10.3390/ph16020249

**Published:** 2023-02-07

**Authors:** Qiao Luo, Ni Shao, Ai-Chen Zhang, Chun-Fang Chen, Duo Wang, Liang-Ping Luo, Ze-Yu Xiao

**Affiliations:** The Guangzhou Key Laboratory of Molecular and Functional Imaging for Clinical Translation, The First Affiliated Hospital of Jinan University, Guangzhou 510632, China

**Keywords:** nanozymes, molecular imaging, magnetic resonance imaging, photoacoustic imaging, positron emission tomography, multimodal imaging

## Abstract

New nanotechnologies for imaging molecules are widely being applied to visualize the expression of specific molecules (e.g., ions, biomarkers) for disease diagnosis. Among various nanoplatforms, nanozymes, which exhibit enzyme-like catalytic activities in vivo, have gained tremendously increasing attention in molecular imaging due to their unique properties such as diverse enzyme-mimicking activities, excellent biocompatibility, ease of surface tenability, and low cost. In addition, by integrating different nanoparticles with superparamagnetic, photoacoustic, fluorescence, and photothermal properties, the nanoenzymes are able to increase the imaging sensitivity and accuracy for better understanding the complexity and the biological process of disease. Moreover, these functions encourage the utilization of nanozymes as therapeutic agents to assist in treatment. In this review, we focus on the applications of nanozymes in molecular imaging and discuss the use of peroxidase (POD), oxidase (OXD), catalase (CAT), and superoxide dismutase (SOD) with different imaging modalities. Further, the applications of nanozymes for cancer treatment, bacterial infection, and inflammation image-guided therapy are discussed. Overall, this review aims to provide a complete reference for research in the interdisciplinary fields of nanotechnology and molecular imaging to promote the advancement and clinical translation of novel biomimetic nanozymes.

## 1. Introduction

As an emerging discipline at the intersection of molecular biology and traditional medical imaging, molecular imaging (MI) promotes the development of precision medicine by visualizing specific molecular and cellular targets related to disease diagnosis and treatment [1,2,3]. Various imaging modalities, including positron emission tomography (PET), single-photon emission computed tomography (SPECT), optical imaging, magnetic resonance imaging (MRI), computed tomography (CT), and ultrasound imaging, have been successfully applied in the field of molecular imaging. To obtain precise MI in vivo, highly sensitive and specific molecular targeted imaging probes have aroused researchers’ interest [4,5]. The ideal molecular probe can facilitate MI with enhanced metabolic stability, favorable pharmacokinetics, improved binding affinity and selectivity, better imaging ability as well as biosafety [6,7,8]. The determination and analysis of these parameters are crucial for the discovery, treatment, monitoring, and prognosis of early diseases [9,10]. Therefore, molecular probes are an indispensable part of a precise MI.

To date, connecting diverse imaging components (radioisotopes, fluorophores, nanoparticles) and targeted ligands (small molecules, peptides, proteins, antibodies, cells) has led to the development of imaging probes suitable for various imaging models [9,11,12,13]. These imaging probes have shown improved imaging performance to a certain extent. However, their development still faces some obstacles, such as the low affinity of molecular imaging probes, low efficiency of molecular probe transcellular transport, insufficient intracellular delivery, and low concentration of the probes in the targeted tissues, leading to inaccurate molecular imaging [14,15].

The interdisciplinary nature of nanotechnology has improved the above problems [16,17]. Nanoprobes have unprecedented potential in personalized medicine, affording many advantages, including dramatic signal amplification, enhanced affinity and specificity, and bypassing of biological barriers [18,19]. Researchers have developed various nanoprobes with specific enzyme-like activities based on the inherent physiochemical properties of materials such as metal oxides [20,21]. In addition to metal oxides, nanozymes can also be composed of metals, metal-organic frameworks, semiconductors, biomimetic materials, and carbon [22,23]. These artificial enzymes based on nanomaterials (nanozymes) have been widely used in many fields, such as biosensors, immunoassays, neuroprotection, and stem cell growth [24,25]. Notably, some studies have also applied nanozymes in imaging [26]. Benefiting from the inherent characteristics of nanozymes (such as the magnetism of Fe, the X-ray absorptivity of Zr, and the optics of Au), MRI, CT, and optical imaging can be used to track the behavior of nanozymes in vivo [27,28,29]. Taking advantage of the catalytic properties of nanozymes, scientists can utilize fluorescent products for imaging [30]. Some studies have also suggested that nanozymes can improve imaging sensitivity [31,32,33].

Despite the significant advancements in nanozymes for MI applications, few studies have systematically reviewed these novel emerging nanomaterials. Hence, in this review, nanozymes, including those with peroxidase (POD)-, oxidase (OXD)-, catalase (CAT)-, and superoxide dismutase (SOD)-like properties, are collectively summarized and their applications in MI, such as with MRI, PET, etc., is comprehensively discussed (Figure 1). In addition, the current challenges and future directions to exploit and develop nanozymes for disease imaging and diagnosis are included. Overall, this review aims to provide a complete reference for research in the interdisciplinary fields of nanotechnology and molecular imaging to better promote the advancement and clinical translation of novel biomimetic nanozymes.

## 2. Classification of Nanozymes

In general, nanozymes can be roughly divided into peroxidase (POD), oxidase (OXD), catalase (CAT), and superoxide dismutase (SOD) based on the substrates they catalyze [34,35,36]. Although various nanomaterials with different properties and physiological functions have been exploited, most of those that are being used for cancer diagnosis and theranostics may be classified as oxidoreductase nanozymes [37,38].

### 2.1. POD-like Nanozymes

PODs usually catalyze substrate oxidation by consuming H_2_O_2_ or organic peroxides [39]. Heme iron protein is the main component of most natural PODs [40]. Inspired by this, researchers have designed and developed various iron-based nanomaterials exhibiting POD-like catalytic activity [41]. For instance, the pioneering work by Gao and coworkers in 2007 indicated that Fe_3_O_4_ nanoparticles possessed intrinsic catalytic activity toward classical POD substrates, including 3,3,5,5-tetramethylbenzidine (TMB), diazo-aminobenzene (DAB), and o-phenylenediamine (OPD) [42]. Later, more iron-containing materials, such as iron–sulfur compounds [41] and Prussian blue [43,44], were proven to exhibit POD catalytic activity [45,46,47]. From this discovery, transition metal elements such as Au, Ag, Pt, and Pd together with Fe can thus be exploited as POD-like nanozymes for diagnosis, therapeutics, and biosensing [48,49,50]. Moreover, metal-organic frameworks (MOFs) consisting of coordinating ions or metal clusters (such as copper and iron) of organic ligands are also emerging as next-generation nanozymes considering that the type and contents of metal components in the structure can be tuned to meet the different demands of various imaging technologies [51]. In addition, carbon-based nanomaterials are another kind of peroxidase-active nanomaterial with great significance [52].

Metals and metal oxides can have CAT-like and POD-like activities. According to research, enzyme activity will be dominated by pH or temperature. Specifically, metal nanomaterials display POD-like activity under acidic conditions. In contrast, CAT-like activity can be observed in metal nanomaterials in an alkaline environment [53]. The tumor microenvironment (TME) provides a favorable environment for tumor progression due to its unique characteristics, such as acidity, hypoxia, inflammation, and excessive hydrogen peroxide generation [54]. Accordingly, the acidic conditions of the TME have become a prominent feature in the activation of nanozyme oxidases [55]. Subsequently, the TME will play a catalytic role in decomposing H_2_O_2_ into O_2_ or reactive oxygen species (ROS) at the tumor site, relieving hypoxia or enhancing the effectiveness of tumor treatment, including photodynamic therapy (PDT), sonodynamic therapy (SDT), and chemodynamic therapy (CDT) [56,57]. For example, the microenvironment can activate the composite nanomaterial Fe_3_O_4_@Carbon@Ptchlorin Ce_6_ (MCPtCe_6_) through a POD-like catalytic process and extend PDT/PTT in acidic and H_2_O_2_-rich microenvironments [58]. This nanomaterial is a multifunctional tumor theranostic agent that can be used for MRI, PTT, PDT, and catalytic therapy. MCPtCe_6_ simultaneously displayed improved photothermal performance and ROS-generating capacity due to the synergistic effects of the four components (PtNPs, Fe_3_O_4_ NPs, Ce_6_, and the carbon shell) of the yolk–shell structure.

### 2.2. OXD-like Nanozymes

Natural oxidases can catalyze the oxidation of a substrate into oxidized products and H_2_O/H_2_O_2_/O_2_ with the assistance of molecular oxygen (or other oxidizing reagents) [59]. Several nanomaterials have been reported to act as oxidases, such as Cu-containing nanoparticles, Pt-based nanoparticles, and manganese dioxide [60].

In Rossi’s study, Au NPs could be used as a GOx (glucose oxidase) mimic to oxidize glucose into H_2_O_2_ and glucose delta-lactone (GDL) [61]. Since then, researchers have begun to explore the potential of Au NP enzyme-like activity in cancer treatment (such as hunger treatment and chemotherapy) [62]. Nanozymes with OXD properties can consume O_2_ to produce ROS, considerably meeting the requirements for tumor treatment. For example, Gao’s research has proven that dendritic mesoporous silica NPs loaded with ultrasmall Au and Fe_3_O_4_ NPs, inorganic nanozymes with multiple enzyme activities, achieve high efficacy and excellent biosafety [63]. Based on the H_2_O_2_ concentration in the tumor, AuNPs explicitly oxidize glucose to gluconic acid and H_2_O_2_. Afterward, POD mimics the Fe_3_O_4_ NPs to catalyze H_2_O_2_ to release ROS (•OH), and tumor cells die via typical Fenton-catalyzed reactions.

Recently, Yuan and coworkers constructed a TME-responsive nanozyme, DOX@HMSN/Mn_3_O_4_(R) [64]. This nanozyme shows activities similar to OXD and CAT, alleviating hypoxic conditions in the TME and reversing the Mn^3+^/Mn^2+^ transformation by TME via in situ triggering, thus disrupting the excessive production of intrinsic redox steady-state catalysis (ROS). Under radiotherapy, high-energy X-rays can stimulate the outer electrons in the nanozymes to form photoelectrons that participate in an OXD-like enzyme reaction, thus enhancing the accumulation of ROS and the effect of radiotherapy/chemotherapy.

### 2.3. CAT-like Nanozymes

CAT can decompose H_2_O_2_ into H_2_O and O_2_ [65,66,67]. Many nanomaterials (e.g., metals and metal oxides) display CAT-like activity [68,69,70]. As mentioned earlier, these reported nanomaterials have CAT-like and POD-like activity, and pH or temperature will determine which type of activity dominates. Recently, Pt and Pd nanomaterials have attracted much interest due to their excellent simulation of CAT activity, near-infrared (NIR) fluorescence characteristics, and photothermal conversion ability [71]. Sun et al. took advantage of the catalase activity of Pt and Pd to design the nanoplatform Pd@Pt-T790 [72]. This nanoplatform was successfully applied to eradicate myositis induced by methicillin-resistant *Staphylococcus aureus* (MRSA). Nanozymes can increase ultrasound-driven dynamic therapy by catalyzing the generation of oxygen. This nanosystem skillfully utilizes the convertible ultrasonic enzyme activity to ensure significant oxygen accumulation at the infected site.

### 2.4. SOD-like Nanozymes

SOD catalyzes the conversion of superoxide free radicals (O_2_·) to O_2_ and H_2_O_2_ [73]. Nanometer cerium is one of the earliest reported nanomaterials with SOD-like activity [74,75]. CeO_2_ nanoparticles have been widely recognized as candidates for superoxide dismutase [76]. Owing to the existence of the mixed valence states Ce^3+^ and Ce^4+^ as well as the presence of oxygen vacancies, CeO_2_ shows a variety of enzyme-like catalytic activities, including those that are POD-like, CAT-like, and SOD-like [75]. For example, a hollow mesoporous Mn/Zr-codoped CeO_2_ tandem nanozyme (PHMZCO-AT) regulates various enzyme activities and disrupts H_2_O_2_ homeostasis, ultimately achieving the therapeutic purpose of CDT [77]. Due to paramagnetic Mn^2+^ and the high atomic number of Zr, *T*_1_-weighted MRI and high contrast X-ray computed tomography (CT) imaging of PHMZCO-AT nanozymes are feasible.

## 3. Nanozymes for Molecular Imaging

Early personalized imaging and diagnosis are crucial for effective cancer treatment with a better prognosis [78]. However, accurately detecting small lesions early and distinguishing and identifying benign and malignant tissues with high sensitivity and specificity are significant obstacles [79]. Therefore, it is necessary and urgent to develop suitable contrast agents with high sensitivity and affinity for malignant tissues for molecular imaging. Currently, many researchers are devoted to improving the detection sensitivity of nanozymes and optimizing the design of nanozyme-like catalytic activity, including size adjustment, composition, heteroatom doping, and specific surface modifications [80,81,82,83,84]. Nanozymes recognize some specific TME biomarkers to improve the diagnostic efficacy of early cancer, such as abnormal fluctuations and expression of metal ions and miRNA [85,86]. Unlike the former, nanozymes for molecular imaging have therapeutic significance in compliance with pharmacokinetics [87,88]. The distribution and activity of nanozymes are tracked with molecular imaging in vivo. Nanozymes aggregate specifically in the TME and perform different enzyme activities to participate in catalytic biochemical reactions, which can resolve the complexity and dynamic variability of the TME and cancer treatment [89,90]. In addition, nanozymes integrate diagnosis with imaging (including MRI, CT, near-infrared (NIR) imaging, and PAI) and treatment (including PTT, PDT, and CDT) [91,92,93,94,95] to provide not only a personalized diagnosis but also a constructive direction for integrated diagnosis and treatment [96]. Considering the different basic principles of each imaging mode, we will focus on typical examples and the latest progress in molecular imaging of each mode’s nanozymes in this section. Characteristic examples of nanozymes with diverse catalytic activities in different imaging modalities are summarized in Table 1.

### 3.1. Magnetic Resonance Imaging

MRI is one of the most effective imaging tools in modern medical imaging owing to its high spatial resolution, excellent soft tissue contrast, imaging in any direction, and low damage to tissues [117,118]. It is widely used in biomedicine, such as central nervous system imaging or cardiac and kidney function evaluation [119,120]. Although MRI is currently the leading imaging technique to detect soft tissues, it is still difficult to discriminate between benign and malignant tissues due to the negligible differences caused by the long relaxation time of water protons [121,122]. Therefore, researchers introduced contrast agents to accelerate the proton relaxation rate to solve this problem, thereby increasing the contrast [123,124]. Notably, with the help of contrast medium, the sensitivity and specificity of MRI in disease diagnosis have been improved [125]. MRI contrast agents can be divided into *T*_1_—weighted (positive) and *T*_2_—weighted (negative) agents according to their effects on longitudinal *T*_1_ or transverse *T*_2_ relaxation [126]. Fast *T*_1_-weighted imaging shows bright contrast in MRI, while the opposite *T*_2_-weighted imaging shows dark contrast [127]. The most commonly used MRI contrast agents are paramagnetic gadolinium complexes, such as gadolinium diethylene triaminepenta acetate (Gd-DTPA) [128,129]. However, due to the relatively low relaxation characteristics of gadolinium-based contrast agents, high doses of contrast agents are needed, which may lead to serious side effects, such as nephrogenic systemic fibrosis (NSF) [130,131]. In addition, due to its inherently short cycle time, it is more difficult to obtain high-resolution images [132]. Even though conjugated macromolecules can overcome this problem with gadolinium chelates, live ions may increase the possibility of side effects [133,134]. Therefore, it is vital to develop new MRI contrast agents that can improve the imaging sensitivity and accuracy of disease diagnosis.

Since the discovery of the POD-like activity of Fe_3_O_4_, an increasing number of iron-based nanozymes (including iron-based nanomaterials and their composites) have attracted extensive attention [135]. Because of the superparamagnetism of Fe_3_O_4_ nanoparticles (Fe_3_O_4_ NPs), the relaxation time of the surrounding protons is shortened, thus providing the possibility of employing this material as *T*_2_—weighted MRI agent media [97,136]. However, owing to the inherent dark signal of *T*_2_—weighted MRI, it is impossible to accurately distinguish tumors from other low-signal areas, such as calcifications, metal deposits, or hemorrhage [98,137]. Researchers found that when the size of an Fe_3_O_4_ NP is less than 5 nm, it works with *T*_1_—weighted MRI because the reduced magnetic moment will strongly inhibit the *T*_2_ effect [138], which provides the possibility of preparing responsive *T*_2_*/T*_1_ switched MRI contrast agents [139,140,141]. Benefiting from their catalytic activity similar to that of peroxidase, Fe_3_O_4_ NPs can promote iron-mediated apoptosis of tumor cells through the Fenton reaction for cancer treatment [142,143,144].

Yu et al. prepared a *T*_1_/*T*_2_ convertible MRI-guided cancer therapeutic agent based on ROS generated by Fe_5_C_2_@Fe_3_O_4_ NPs (Figure 1) [99]. In endogenous tumor environments, Fe_5_C_2_@Fe_3_O_4_ NPs release ferrous ions due to the low pH, which is disproportionate to the excessive H_2_O_2_ at the tumor site, and generate •OH radicals through the Fenton reaction, thus effectively acting against tumors. The high magnetic properties of Fe_5_C_2_@Fe_3_O_4_ NPs make them capable of *T*_2_—weighted MRI. Moreover, the ionization of these NPs in an acidic environment improves the *T*_1_ signal. This switchable *T*_2_/*T*_1_ process visualizes ferrous ion release and ROS generation for cancer therapy. Fe_5_C_2_@Fe_3_O_4_ NPs have high efficiency and tumor specificity, showing great therapeutic potential in the TME under the guidance of MRI. Further modification of Fe_3_O_4_ NPs with photothermal agents or sonosensitizers may provide multimodal cancer treatment with photothermal therapy (PTT) or photodynamic therapy (PDT). For example, altering magnetic Fe_3_O_4_ NPs with the classical photothermal agent (AuNPs), in addition to MRI and magnetic targeting functions, can bring additional PTT capabilities and improve the accuracy of Fe_3_O_4_ NPs to diagnose and treat tumors. As a result, Fe_3_O_4_ NPs have become a novel nanoplatform for personalized tumor theranostics.

Likewise, as another transition metal element, manganese can also be exploited as an MRI contrast agent [145,146,147]. Manganese oxide nanoparticles (MONs) are potential *T*_1_—weighted MR contrast agent candidates. Mechanistically, the five unpaired electrons in the three-dimensional orbital of this ion can produce significant magnetic moments and cause the relaxation of nearby water protons [148,149]. Studies have shown that MONs can mimic the characteristics of POD, CAT, and SOD. Among various MONs, MnO_2_ nanostructures have attracted considerable attention as stimuli-responsive and biodegradable materials [150,151]. In the TME, MnO_2_ can react with GSH, H^+^ ions, and H_2_O_2_ to generate Mn^2+^ ions, playing multiple roles in tumor diagnosis and treatment. MnO_2_ can also be an MR probe that is specifically activated in the TME to accurately target tumors. Taking advantage of their CAT-like activity, MnO_2_ nanostructures can decompose excessive H_2_O_2_ into O_2_ in situ, relieving hypoxia and enhancing other treatments. As a drug vehicle, MnO_2_ nanoshells can also be decomposed by GSH and H^+^ ions in the tumor microenvironment. Thus, the loaded chemotherapeutic drugs can be unleashed in response to achieve precise treatment [152,153,154,155]. MnO_2_ nanostructures are regarded as CDT agents because they catalyze H_2_O_2_ in the TME to produce •OH via Fenton-like reactions.

For example, Huang et al. designed the multifunctional bionic thermal nanoplatform Au@Pd@MnO_2_ (APMN NPs) and further coated it with a natural killer cell membrane to form a core-shell structure (Figure 2) [100]. The Au@Pd nucleus was determined to have a high-temperature effect and nanozyme catalytic activity under near-infrared (NIR) activation. The mesoporous MnO_2_ shell provides MR imaging and high drug-loading ability and endows the nanoplatform with various enzyme-like activities, such as those of POD and XOD. In addition, NK-92 cell membrane camouflage endows APMN NPs with enhanced tumor-targeting ability, immune escape functions, and membrane-protein-mediated tumor uptake characteristics. The TME-responsive MR imaging and drug-release characteristics make APMN NPs an intelligent integrated nanoplatform. The developed NPs showed high therapeutic efficacy in MCF-7 tumor-bearing mice for antitumor therapy and biosafety in histological and blood biochemical experiments.

Metal-organic frameworks (MOFs) containing transition elements are also emerging as MRI contrast agents [156,157]. Prussian blue (PB), as a vital MOF, is a promising material with unique physical and chemical properties [158]. The carbon-bonded Fe(II) (S = 0) and nitrogen-bonded Fe(III) (S = 5/2) in PB produce five unpaired electron ±C≡N − Fe^3+^ units for each Fe^2+^, which can shorten the relaxation time of protons and enable them to be used as contrast agents for magnetic resonance imaging (MRI) [159]. In recent years, researchers have also found that PBNPs exhibit catalytic properties such as those of POD, CAT, and SOD [160]. PBNPs can potentially treat symptoms caused by ROS due to their multienzyme activities and ROS-clearance abilities [161]. Zhang’s team successfully fabricated multifunctional ultrasmall nanozymes based on PB (PB NZs) [101]. The as-prepared PB NZs exhibited efficient cellular protection from H_2_O_2_—induced damage in vitro because of their excellent multienzyme-mimicking capabilities to remove excessive RONS [e.g., H_2_O_2_, the hydroxyl radical (•OH), the superoxide anion (O_2_•^−^), nitric oxide (NO), and peroxynitrite (ONOO^−^)]. Based on their MR and PA imaging properties, PB NZs accumulate rapidly and efficiently in the kidneys of AKI mice. The ultrasmall size of PB NZs produced negligible systemic toxicity in mice. In addition, serum and biomarker testing, histological staining, and mouse survival studies have demonstrated that ultrasmall PB NZs have good biocompatibility. Consequently, ultrasmall PB NZs have excellent potential as a nanozyme therapy agent to treat AKI.

### 3.2. Photoacoustic Imaging

At present, photoacoustic imaging (PAI) has become a promising noninvasive imaging method that combines spectral selectivity from laser excitation with the high resolution of ultrasonic imaging [162,163]. Sensitive light absorption, high spatial resolution, and imaging depth are the advantages of PAI [164,165]. Compared with ultraviolet and visible light, the interaction of near-infrared light with biological tissues is relatively weak, which is more conducive to clinical diagnosis [166,167]. The near-infrared region includes the first (NIR-Ⅰ, 650–9950 nm) and the second (NIR-II, 1000–1700 nm) [168,169] windows. NIR-II window PAI has apparent advantages in terms of penetration depth and signal-to-noise ratio (SNR) [169,170]; thus, it has aroused increasing interest from researchers [171].

In addition to the excitation light source, the photothermal effect based on photothermal transduction agents (PTAs) is also a focus of PAI [164,172]. PTAs absorb light energy and convert it into heat to raise the temperature of the surrounding environment, and then the tissue undergoes temporary thermoelastic expansion to generate ultrasonic waves [173,174]. Eventually, the generated ultrasonic signals are gathered with broadband ultrasonic transducers and transformed into PA images [172]. Generally, PTAs can be used for PTT and PAI [175]. PTT involves increasing the temperature of the surrounding environment to cause cancer cell death through the photothermal effect [176]. Thus, PTAs lay a solid foundation for the integration of diagnosis and treatment [177]. The perfect PTA should have a high photothermal conversion efficiency, an excellent signal-to-noise ratio, and satisfactory tumor accumulation [178]. PTAs can be classified as inorganic materials or organic materials. Inorganic materials include noble metal materials (Au, Ag, Pt, Pd), metal chalcogenide materials, carbon-based nanomaterials (e.g., graphene and carbon nanotubes), and other two-dimensional (2D) materials (such as nanosheets, boron nitride, and graphite carbonitride) [179,180,181]. Typically, inorganic PTAs have higher photothermal conversion efficiency and better stability than organic agents. Regardless, organic PTAs are superior in terms of biodegradability and biocompatibility.

Specifically, after the laser light is absorbed by noble metal PTAs, the electrons are converted from the ground state to the excited state. Finally, they release energy via nonradiative attenuation as heat [102,182]. Additionally, these PTAs can simulate the catalytic activities of POD, OXD, CAT, etc. In the microenvironment, nanozymes have one or more enzyme activities and can effectively solve cancer treatment problems with PTT, PDT, or CDT. As proof, Wang et al. manufactured a combined treatment nanoplatform (Au_2_Pt-PEG-Ce6), with the photosensitizer chlorin e6 (Ce6) covalently linked to convert O_2_ into singlet oxygen (^1^O_2_), which significantly enhanced the stability of ^1^O_2_ and avoided its premature release in complex organisms [103]. This platform has CAT and POD dual catalytic activities, which can generate O_2_ and •OH to enhance PDT and CDT. Additionally, Au_2_Pt-PEG-Ce6 can effectively achieve PDT and PTT under 650 nm and 808 nm laser irradiation. Furthermore, due to the strong absorption in the NIR region, Au2Pt-PEG-Ce6 can serve as a photothermal transition agent for NIR-responsive PTT and provide the possibility of PA and photothermal (PT) imaging. Moreover, the extraordinary X-ray attenuation capabilities of Au and Pt endow Au2Pt-PEG-Ce6 nanomaterials with the ability to be CT imaging contrast agents. Importantly, image-guided tumor therapy has confirmed that Au2PtPEG-Ce6 can significantly inhibit tumor growth without side effects to normal tissues. The imaging results were consistent and the tumor showed an enhanced signal, while only a weak signal was observed in the control group. Notably, the image-guided tumor treatment results demonstrated that Au2PtPEG-Ce6 can effectively inhibit tumor growth with almost no side effects to normal tissues. Additionally, in vivo toxicity experiments showed that Au2Pt-PEG-Ce6 has outstanding biocompatibility, making it promising for biological applications.

Liang and coworkers designed and synthesized the excellent hollow structure Pt CuS Janus (Figure 3) [104]. The hollow interior of CuS supplies a great space for TAPP molecule packing to enforce sonodynamic therapy. Moreover, Pt endowed with the catalytic property of CAT modulates the engagement of O_2_ to alleviate tumor hypoxia and boost sonodynamic-therapy-induced ROS production. Significantly, the CAT-like enzyme activity of Pt can be accelerated by the heat generated under 808 nm laser irradiation to generate more O_2_, thereby enhancing the effect of sonodynamic therapy. Finally, after the temperature-sensitive polymer poly(oligo(ethylene oxide) methacrylic acid co-2-(2-methoxy ethoxy)methacrylate) (p-(OEOMA-co-MEMA)) was covered with Pt-CuS NPs (Pt-CuS-P-TAPP, labeled PCPT), the nanozyme activity and drug-release rate were effectively controlled by temperature. Moreover, the nanosystems, acting as attractive PA imaging agents and NIR thermal imaging agents, were further optimized to guide the therapeutic options for cancer PA imaging. The tumor signal from CT26 tumor-bearing mice was evident and increased to a maximum value at 12 h. Under 808 nm laser irradiation, a rapid increase in tumor temperature via the effective accumulation of PCPT in tumors not only destroys tumor tissue but also ensures its potential clinical transformation due to its high biosafety.

However, PTAs based on precious metals face the problems of high cost and nondegradability, which are complex problems for clinical conversion. Thus, scientists looked for other inorganic materials for use as PTAs. Carbon-based materials with broad optical absorption and reasonable photothermal properties, such as graphene, graphene oxide, and carbon nanotubes, have attracted significant attention. The intrinsic POD-like activity of graphene quantum dot nanozymes (GQDzymes) effectively converts 2,2′-azino-bis (3-ethylbenzothiazoline-6-sulfonic acid) (ABTS) into its oxidized form in the presence of H_2_O_2_. Oxidized ABTS exhibits strong near-infrared (NIR) absorbance, rendering it an ideal contrast agent for PAI. For instance, Ding and coworkers designed a novel H_2_O_2_-responsive exosome-like nanozyme vesicle for PAI detection of nasopharyngeal carcinoma [105]. They developed an approach to construct exosome-like nanozyme vesicles via biomimetic functionalization of GQDzyme/ABTS nanoparticles with natural erythrocyte membranes modified with folate acid. This biomimetic decoration contributes to prolonging circulation time, improving tumor accumulation, and facilitating tumor uptake, which were superior to those exhibited by traditional nonbiological material modification strategies. Notably, due to the POD activity, light absorption, and photothermal properties of GQDzyme, the PA signal is enhanced in the H_2_O_2_-containing tumor environment. In addition, nanovesicles also have good biocompatibility and long-term blood circulation invisibility.

Based on Ding’s research [103], Liu et al. loaded ABTS into the POD-like metal-organic framework (MOF) MIL-100. They coated the frame with polyvinylpyrrolidone (PVP) to design activated ABTS@MIL-100/PVP nanoreactors (AMP NRs) (Figure 4) [106]. The tumor microenvironment can activate these AMP NRs to display their photoacoustic imaging signal and perform photothermal therapy (PTT). In addition, the high levels of H_2_O_2_ in the tumor microenvironment undergoes a reaction to produce hydroxyl radicals and destroy intracellular glutathione (GSH), giving AMP NRs the ability to carry out enhanced chemodynamic therapy (ECDT). After intravenous injection of AMP NRs, 4T1 tumor-bearing mice at different tumor-growth stages (5 and 12 days after inoculation) were imaged, and the data highlighted the excellent ability of the nanoreactor to detect tumors in the early stage. Additionally, AMP NRs completely inhibited cancer growth in 4T1 tumor-bearing mice by satisfactory ECD and PTT. The low toxicity and high biocompatibility of AMP NRs ensure their clinical transformation.

NIR-II PA imaging has been extensively discussed and studied due to its excellent accuracy in diagnosing diseases and the noninvasive antitumor performance of PTT under imaging guidance [183]. Single PAI cannot effectively provide tumor information with distinct spatial resolution and scale. NIR-II fluorescence imaging stands out in optical imaging because of its good spatial resolution, high sensitivity, and negligible background interference. Accordingly, researchers have constructed dual-modal imaging techniques by integrating NIR-II PA and NIR-II FL imaging technology to supply accurate and successful imaging details and guide precise antitumor treatment. For instance, Zheng and coworkers developed a new nanozyme, HSC-2 (Figure 5) [107]. HSC-2 has an adjustable dual-modal NIR-II PA/NIR-II FL imaging ability to guide precise, synergistic, catalytic photothermal therapy. In this system, by adjusting the band gap and surface N-doped carbon content to change the carbon–silicon ratio in the frame, the light absorption and emission capabilities in the NIR-II window were optimized simultaneously. Due to its optimum silicon-carbon ratio, HSC-2 produced strong NIR-II PA absorption and excellent NIR-II fluorescence emission performance. As a dual-modal imaging probe, HSC-2 not only detects deep tumor tissues but can also display comprehensive information on tumors with high resolution. The POD-like property of HSC-2 can transform endogenous H_2_O_2_ in tumor areas into highly toxic ROS (•OH and O_2_•^−^). Additionally, because of its excellent photothermal conversion efficiency and POD-like catalytic ability, HSC-2 generates stable high temperatures and sufficient ROS in tumor tissues to produce antitumor effects.

### 3.3. Other Types of Optical Imaging

Optical imaging is an indispensable component of molecular imaging [184]. In recent years, the benefits of optical imaging, such as high sensitivity, great cost-effectiveness, nonionizing effects, and real-time imaging capabilities, have attracted much attention in various studies of in vivo and in vitro systems [185]. Nevertheless, optical imaging in vivo has many limitations caused by light scattering, intrinsic fluorescence, and absorption by adjacent tissues, water, and lipids [186,187] Hence, some advanced imaging technologies (e.g., fluorescence, bioluminescence, diffusion optical tomography, and optical coherence tomography) have been developed to improve the status quo [188,189,190]. In particular, near-infrared fluorescence (NIRF) imaging is a current method that is widely used for imaging small animals in vivo [191]. This section will focus on NIRF imaging probes.

The primary principle of fluorescence imaging is that external light excites specific fluorophores and a susceptible charge-coupled device camera is used to detect emission [192]. The fluorophores can be endogenous molecules, such as hemoglobin, or exogenous molecules, such as synthetic optical probes (e.g., fluorescein isothiocyanate and rhodamine) [193]. Additionally, because optical probes based on nanozymes were initially designed to be turned off, they can only respond to specific signals when subjected to complex microenvironments [194]. Therefore, this kind of optical probe imaging has the advantages of low background noise and a high signal-to-noise ratio.

In Zhang’s research, smart macrophages loaded with GOx nanoparticles expressed by IR820 macrophages were successfully constructed for synergistic photothermal activation hunger therapy in tumors (Figure 6) [108]. Because macrophages have excellent phagocytosis abilities, H_2_O_2_-sensitive GOx nanozymes were loaded onto the macrophages to load drugs effectively. Second, macrophages carrying therapeutic agents naturally accumulate at the tumor site due to their inherent tumor tropism. This process creates the fluorescence imaging ability obtained by the photosensitizer IR820 to track the transportation process. Finally, the external laser and special TME trigger the spatiotemporal unpacking of the loaded therapeutic agent at the tumor site. Under fluorescence guidance, the GOx enzyme activity is no longer shielded, and starvation treatment begins. At this time, positive feedback is initiated in the tumor microenvironment, and the newly generated H_2_O_2_ further induces the activation of GNPs. In addition, laser-triggered PTT induces apoptosis of tumor cells for local tumor hyperthermia, accelerates the decomposition of H_2_O_2_ into O_2_, and enhances GOx enzyme activity for hunger therapy. Therefore, IRG@RC plays a variety of roles, such as those that promote laser-induced drug release and activation, the response of the tumor microenvironment, and circular amplification characteristics, realizing synergistic PTT and hunger therapy in vitro and in vivo.

The intrinsic catalytic properties of noble metal nanomaterials have been combined with different tumor therapies [109]. In addition, noble metals have optical imaging abilities due to their light stability and strong light signal. Researchers developed an image-guided treatment to overcome the dilemma of tumor chemotherapy tolerance. For example, approximately 100 kinds of NIR-II gold nanoclusters smaller than 3 nm have drawn substantial awareness for their intrinsic FL emission, renal clearance, and high biocompatibility. Along with their FL imaging capability, Au-NCs may also produce toxic ^1^O_2_ under NIR laser irradiation to enhance PDT to obliterate cancer cells and bacteria. Au-NCs exhibit CAT-like qualities to decompose the endogenous hydrogen peroxide (H_2_O_2_) in tumor tissue into O_2_ to relieve tumor hypoxia. Hence, Dan et al. designed a new photosensitizer to improve PDT by utilizing the NIR-II fluorescence property and CAT activity of a gold NC (BSA@Au) [110]. Because of its bright NIR-II fluorescence, BSA@Au can pinpoint the tumor location with a high signal background ratio (SBR = 7.3) in 4T1 tumor-bearing mouse models and conduct efficient PDT. Second, the biological distribution and metabolic pathway of BSA@Au in vivo ensure safety for clinical conversion. Compared with the control group, the survival time of tumor-bearing mice treated with the BSA@Au-based PDT strategy was five times longer. Furthermore, BSA@Au-based PDT can protect against bacterial infections.

### 3.4. Positron Emission Tomography

PET has been widely used in clinical practice for nearly half a century due to its extremely high sensitivity, excellent penetration depth, and signal that can be quantitatively analyzed [195]. The positrons emitted by the accumulated radionuclides in living subjects are annihilated and detected [196]. PET scanners detect isotopes in deep tissues at picomolar concentrations [197]. This unique sensitivity can be used to obtain quantitative information from drugs that have accumulated at target sites [198]. In addition, the amount of nanomaterial needed for PET imaging is minimal, and the substance will not interfere with biological systems and is unlikely to induce toxic reactions [199]. Combined imaging modes, including PET/CT and PET/MRI, have been explored to overcome inherent restrictions, such as the lack of anatomical information and low resolution [200]. The joint development of PET and nanotechnology has mutual benefits [201]. The application of nanozymes in PET is expected to become unprecedented thanks to their unique physical and chemical properties [197]. Furthermore, PET is an excellent choice for scientists to optimize the pharmacokinetics, intratumoral penetration, tumor bioavailability, and cargo-delivery mechanisms of nanoparticulate agents [202,203].

Nanozymes that have become PET probes include iron oxide nanoparticles, manganese oxide nanoparticles, Au nanoparticles, and graphene oxide nanoparticles [204,205]. Gold nanoparticles (Au NPs) are glucose OXD-like nanozymes that can efficiently catalyze glucose into H_2_O_2_ and gluconic acid. However, suppose that a large amount of surfactant is added during the preparation process to stabilize the ultrasmall AuNPs. In this case, the ultrasmall AuNPs cannot be used directly for biomedical applications. Additionally, these small Au NPs are rapidly cleared from systemic circulation by the kidneys. Therefore, an ideal nanocarrier is needed to immobilize ultrasmall Au NPs to ensure their biological safety and catalytic effect in vivo. Li et al. reported the in situ design of polymeric hollow mesoporous organosilicon nanoparticles (HMONs) to immobilize ultrasmall Au NPs and provide CDT with self-supplied H_2_O_2_ (Figure 7) [111]. Additionally, a Cu2^+^ tannic acid complex was deposited on the surface of the HMONs to initiate a Fenton-like reaction and convert the self-provided H_2_O_2_ into •OH, a highly toxic ROS. Last, collagenase (Col), which can degrade collagen I fibers in the extracellular matrix (ECM), was loaded into the biocatalytic nanoreactor to increase the permeability of the HMONs and O_2_ to enhance PDT. To determine the targeting effects of the nanomaterial HMON-Au-Col@Cu-TA-PVP in vivo, researchers performed PET imaging and NIR-II imaging. The results showed that the nanoprobe was clustered at the tumor site and reached its peak value in 24 h. The determined biological distribution in vivo lays a foundation for the clinical transformation of this nanomaterial. After CDT, approximately 80% of the pancreatic cancer xenograft tumor model animals experienced rapid tumor regression and long-term tumor-free survival.

Recently, Yuan et al. radiolabeled FHNPs using a thermally induced radiolabeling strategy (PET ^89^Zr^4+^ or ^64^Cu^2+^, SPECT ^111^In^3+^) for PET or single photon emission computed tomography (SPECT) imaging [112]. Feraheme nanoparticles (FHNPs), a very small superparamagnetic iron oxide, have been widely used in the treatment of iron anemia (due to its slow release of ionic iron in acidic environments) and MRI contrast agents. In addition, injected FHNPs are internalized by monocytes and serve as MRI biomarkers for the pathological accumulation of monocytes in diseases. PET/SPECT has essential benefits for FHNP imaging over MRI due to the low iron content and the increased ability to quantify the NP concentration in tissues. Radioactively labeling FHNPs maintains the physical and biological characteristics of the NPs. ^89^Zr-FHNPs showed slow accumulation in mouse lymph nodes and inflammatory sites, which were similar to the MRI uptake results. Finally, by comparing the biological distribution of radiolabeled FH with that of ^89^Zr^4+^ or ^64^Cu^2+^ cations, the in vivo stability of FH can be determined. Radionuclide-labeled FHNPs are functional radioactive nanomaterials for lymph node mapping and PET imaging of inflammation and related inflammatory diseases.

### 3.5. Multimodal Imaging

Single imaging technologies cannot meet the current needs of personalized diagnosis and treatment; however, multimodal imaging can be used to compensate for the deficiencies [206]. Distinct imaging technologies combine multiple imaging modalities to overcome the intrinsic drawbacks of individual imaging technologies and convey more comprehensive and specific imaging information [207,208]. Until now, PET/CT has been the only multimodal molecular imaging strategy demonstrated to be clinically transformable. Other bimodal modalities, such as MRI-optics, MRI-PET/SPECT, MRI-CT, MRI-MPI, MRI-MMUI, and MRI-MPA, are still in the preclinical stage and should be further studied. With the continuous development of multifunctional imaging probes, the robust expansion of multimodal imaging is inevitable [209,210,211,212]. Nanozymes with various composite materials have high imaging efficiency under endogenous and exogenous stimuli and can enhance treatment [113,114,213].

In Gong’s research, a unique, favorably efficient, and stable POD-mimicking nanozyme, FeWOX, was constructed (Figure 8) [115]. The flaky structure with considerable exposure of iron atoms and positively distributed oxygen vacancies (catalytic sites) on its surface endow FeWOX nanozymes with significant activity to decompose H_2_O_2_ into hydroxyl radicals (•OH). In addition, based on its effective enzyme activity and sheet structure, the FeWOX nanozyme was used to construct a ratiometric nanoprobe (FeTIR) based on H_2_O_2_-activated nanozymes, which coanchored TMB and IR780 for PA imaging and displayed high sensitivity and stability. The subcutaneous 4T1 xenotransplantation tumor model and LPS-induced inflammation model proved that the obtained FeTIR nanoprobe had excellent endogenous H_2_O_2_ ratio measurement PA imaging performance in vivo after local injection. Second, FeTIR mainly works because the IR780 dye becomes a fluorescent probe. Moreover, the CT imaging ability comes from vital X-ray attenuation of tungsten (W). Finally, FeTIR can also be used as an MR T_2_-weighted imaging contrast agent due to the presence of iron. This probe integrates multiple imaging functions into a single nanoplatform. Different imaging modes can verify the specificity of the probe to target tumors and compensate for the inherent defects of the other’s imaging methods.

Currently, the types of nanozymes are limited, their efficiency is usually impaired, and their presence is not sustainable in the tumor microenvironment (TME) [214,215]. For example, some nanozymes mimic POD and CAT to assist with CDT or PDT, but the supply of H_2_O_2_ in the TME is limited and not maintainable and their efficiency is usually compromised [216]. Even if combined treatment with other therapies is carried out, the use of nanozymes is not practical because of the strong side effects, high cost, and low efficiency [217]. For instance, Cao et al. designed unprecedented “integrated” Fe_3_O_4_/Ag/Bi_2_MoO_6_ nanoparticles (FAB-NPs), which are highly efficient and specific and have few side effects [116]. FAB-NPs have high POD, CAT, SOD, glutathione oxidase (GSHOD), and photodynamic activity. The doping of Fe_3_O_4_ and AgNPs gives Bi2MoO6 strong NIR-II absorption, enhanced photocatalytic activity, and ferromagnetic and photothermal effects. These properties enable MR, PA, and PT imaging to guide nanocatalytic therapy. The multimodal imaging results showed that FAB-NPs accumulated specifically at the 4T1 breast cancer tumor site and increased with time. After laser irradiation twice a day, the tumor disappeared entirely within 16 days. The blood biochemical and complete blood group analyses after treatment showed no abnormalities. In vitro and in vivo experiments proved that this “integrated” nanoparticle system can achieve synergistic CDT, PDT, and PTT. Therefore, the self-enhancement from cascade nanocatalytic reactions and various nanozyme activities can become a new direction of nanozyme research.

## 4. Conclusions and Outlook

As a unique functional material, nanozymes have been extensively explored and have shown remarkable benefits and potential applications in the biomedical field, mainly diagnosis and treatment. Nanozyme-integrated diagnosis and therapy have unprecedented potential to achieve precise treatment of glioblastoma, breast cancer, hepatocarcinoma, colon cancer, pancreatic ductal adenocarcinoma, cervical cancer, and nasopharyngeal carcinoma. Due to their inherent physical properties, nanozymes can react to external stimuli, which is conducive to regulating their biological behaviors and processes in vivo (e.g., targeted imaging, treatment, and drug release). Adjustment of the metal ion proportion, changes in enzyme catalytic activity, and other types of structural adjustments can specifically enhance imaging sensitivity. As rising stars, nanozymes have made precision medicine a reality.

Nevertheless, flourishing nanozymes still experience many challenges.

Poor dispersion, uncontrollable precipitation after surface modification, limited types of catalytic activity, poor substrate selectivity, and potential nanotoxicity due to the drawbacks of current preparation technologies and methods.Systemic toxicity derived from the overaccumulation of metal elements; for instance, the direct toxic reaction caused by gold metal overaccumulation after long-term application and the indirect toxic response, such as ion-induced oxidative stress, DNA damage, and normal tissue apoptosis. Reducing potential adverse side effects and improving the accuracy and efficacy of diagnosis and treatment are significant problems that must be solved for the viable and sustainable development of metal nanozymes.There is a lack of a comprehensive monitoring and evaluation system for nanozymes after administration in living systems, such as histocompatibility, blood compatibility, immunogenicity, biological distribution, cytotoxicity, in vivo uptake, and metabolism. These factors affect the effective clinical conversion of nanozymes.

## Data Availability

Not applicable.

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
