# Peer review of "Smart Biomimetic Nanozymes for Precise Molecular Imaging: Application and Challenges"

_pharmaceuticals, 2023, doi:10.3390/ph16020249_

Round 1

Reviewer 1 Report

The authors are describing an interesting and complete review about nanozyme and MI. It was a pleasure to read it as a reviewer. As the authors claim, the subject necessite always update due to the several advances in this field. 

I would have few comments and suggestions:

-Ln36, the sentence about the ideal molecular probe is not clear. I would suggest to spend more time on it. The pharmacokinetic is important also in the probe design, doi: 10.2174/156802610791384225,  DOI: 10.1016/j.pharmthera.2020.107786, https://doi.org/10.3389/fmed.2022.812270). This include the fast elimination within the bloodstream while maintening a good uptake in the tumor. This part is important for the rest of the review. 

- Some references on nanozymes are missing:

    ¤ https://doi.org/10.1021/acs.accounts.9b00140

    ¤ https://doi.org/10.1186/s12951-022-01295-y

- Scheme 1 --> Some nanozyme structure would be welcome. The classification of the nanozymes is clear and well described. I would add some chemical structure to have a clear representation.

-  In the imaging part, I would separate or at least distinguish in one/two sentences the use of molecular imaging to follow the distribution and the activity of the nanozyme design for therapy (to follow the drug distribution) and the specific use of nanozyme to detect some specific TME biomarkers. It is not the same question since for the second, the imaging probe based on nanozyme would not have a therapeutic interest.

Author Response

Dear reviewers:

         We sincerely appreciate your thoughtful and professional comments on our manuscript. We have adopted virtually all the suggestions and revised the manuscript accordingly. We hope that the revised manuscript has been improved to the level of the reviewers’ satisfaction. We also hope that you will find it suitable for publication in Pharmaceuticals.

Reviewer #1: The authors are describing an interesting and complete review of nanozyme and MI. It was a pleasure to read it as a reviewer. As the authors claim, the subject necessite always update due to the several advances in this field.

Response: Thanks for your comments and constructive suggestion. Over the last few days, we have made a series of revisions to the manuscript based on your opinion. We expect that the revision will meet the high publication standards of journals and reviewers.

Point 1. Ln36, the sentence about the ideal molecular probe is not clear. I would suggest to spend more time on it. The pharmacokinetic is important also in the probe design, doi:10.2174/156802610791384225,DOI:10.1016/j.pharmthera.2020.107786, https://doi.org/10.3389/fmed.2022.812270). This include the fast elimination within the bloodstream while maintening a good uptake in the tumor. This part is important for the rest of the review.

Response 1: Thank you very much for your valuable comments. We have revised the description of ideal molecular probes in the revised manuscript with yellow highlighting. (Line37-39).

Point 2. Some references on nanozymes are missing:

-https://doi.org/10.1021/acs.accounts.9b00140,

-https://doi.org/10.1186/s12951-022-01295-y.

Response 2: Thank you very much for your valuable comments. We have added the above references to our revised manuscript (References 39,90).

Point 3. Scheme 1 --> Some nanozyme structure would be welcome. The classification of the nanozymes is clear and well described. I would add some chemical structure to have a clear representation.

Response 3:We appreciate the reviewer’s suggestion and comments. We supplemented the chemical structure in Scheme 1.

Point 4. In the imaging part, I would separate or at least distinguish in one/two sentences the use of molecular imaging to follow the distribution and the activity of the nanozyme design for therapy (to follow the drug distribution) and the specific use of nanozyme to detect some specific TME biomarkers. It is not the same question since for the second, the imaging probe based on nanozyme would not have a therapeutic interest.

Response 4: Thank you for the nice comments. We have added the distinction of nanozymes for molecular imaging and detection of specific TME biomarkers. Relevant discussions have been included in our revised manuscript with yellow highlighting (line182-189).

Sincerely,

Zeyu Xiao, Ph.D, M.D.

Reviewer 2 Report

The manuscript of Luo and al. about the selection of nanozymes for molecular imaging and theranostics is well-designed and organised. The physicochemical properties of nanozymes have been correctly analysed and the main issue about the use for molecular imaging correctly reviewed. The manuscript does not require further revision and can be considered suitable for publication in Pharmaceuticals.

Author Response

Dear reviewer,

We sincerely appreciate your thoughtful and professional comments on our manuscript entitled "Smart Biomimetic Nanozymes for Precise Molecular Imaging: application and challenges”. We are grateful for your effort in reviewing our manuscript and your positive feedback. The summary of our work as written is precise. Thank you again.

Sincerely,

Zeyu Xiao, Ph.D, M.D.